# Effectiveness, Flexibility and Safety of Switching IVF to IVM as a Rescue Strategy in Unexpected Poor Ovarian Response for PCOS Infertility Patients

**DOI:** 10.3390/jcm12051978

**Published:** 2023-03-02

**Authors:** Wei Guo, Xiaoying Zheng, Danni Zheng, Zi Yang, Shuo Yang, Rui Yang, Rong Li, Jie Qiao

**Affiliations:** 1Centre for Reproductive Medicine, Department of Obstetrics and Gynecology, Peking University Third Hospital, Beijing 100191, China; 2National Clinical Research Centre for Obstetrics and Gynecology, Beijing 100191, China; 3Key Laboratory of Assisted Reproduction, Peking University, Ministry of Education, Beijing 100191, China; 4Beijing Key Laboratory of Reproductive Endocrinology and Assisted Reproductive Technology, Beijing 100191, China; 5Research Units of Comprehensive Diagnosis and Treatment of Oocyte Maturation Arrest, Chinese Academy of Medical Sciences, Beijing 100006, China

**Keywords:** in vitro maturation (IVM), in vitro fertilization (IVF), ovarian stimulation, polycystic ovary syndrome (PCOS), unexpected poor ovarian response (UPOR)

## Abstract

Background: According to the latest practice committee document, in vitro maturation (IVM) is a simple and safe procedure, especially in patients with polycystic ovary syndrome (PCOS). Does switching from in vitro fertilization (IVF) to IVM (IVF/M) help as a rescue infertility treatment for PCOS patients with an unexpected poor ovarian response (UPOR) tendency? Methods: This retrospective cohort study included 531 women with PCOS who had undergone 588 natural IVM cycles or had switched to IVF/M cycles from 2008 to 2017. Natural IVM was performed in 377 cycles, and switching IVF/M was performed in 211 cycles. The primary outcome measure was the cumulative live birth rates (cLBRs), and the secondary outcomes included laboratory and clinical outcomes, maternal safety, and obstetric and perinatal complications. Results: No significant difference was found in the cLBRs between the natural IVM and switching IVF/M groups (23.6% vs. 17.4%, *p* = 0.05). Meanwhile, the natural IVM group had a higher cumulative clinical pregnancy rate (36.0% vs. 26.0%, *p* = 0.01), and a decrease in the number of oocytes was obtained in the switching IVF/M group (13.5 vs. 12.0, *p* < 0.01). The number of good quality embryos in the natural IVM group was 2.2 ± 2.5, and 2.1 ± 2.3 (*p* = 0.64) in the switching IVF/M group. No statistically significant differences were observed in the number of 2 pronuclear (2PN) and available embryos. Ovarian hyperstimulation syndrome (OHSS) did not occur in the switching IVF/M and natural IVM groups, indicating a highly favorable outcome. Conclusion: In PCOS infertile women with UPOR, timely switching IVF/M is a viable option that markedly reduces the canceled cycle, results in reasonable oocyte retrieval, and leads to live births.

## 1. Introduction

In 2021, the American Society for Reproductive Medicine (ASRM) presented a new practice committee document stating that in vitro maturation (IVM) should no longer be considered an experimental technique, which is an improvement from the document that was published in 2013 [1,2,3]. Researchers have implemented various protocols to improve the outcome of IVM, which has reached the development of human-assisted reproductive technology, where its clinical utilization will both broaden and enrich the hopes of practitioners and patients in the decades ahead [4,5,6,7,8,9,10].

Polycystic ovary syndrome (PCOS) may be the most frequently encountered endocrinopathy in women of reproductive age, for whom in vitro fertilization (IVF) is an effective treatment [11,12,13]. However, patients with PCOS generally show outcomes with a larger variance in IVF treatments compared to normovulatory infertile patients, who have been classified as “poor” and “high” responders [14,15]. High responders have an exaggerated response to gonadotropin (Gn) stimulation and a high risk of ovarian hyperstimulation syndrome (OHSS). In the IVM protocol, a small dose of Gn or even zero Gn treatment avoids OHSS. In contrast, poor responders present an inadequate response to hormonal stimulation, even with the use of large Gn doses, and often have only a few leading (<3) or absent follicles, with low serum estrogen (E2) levels [16]. There is no specific treatment for these patients, and most IVF cycles have been discontinued.

Chian et al. [17] first demonstrated the pregnancies and live births that resulted from IVF of mature oocytes retrieved from dominant follicles in a natural cycle combined with IVM of immature oocytes retrieved from small follicles, and indicated that the natural cycle IVF/M was an attractive choice for infertility treatment for various reasons. In 2022, Zhao et al. [18] discovered that switching from controlled ovarian stimulation (COS) IVF to IVM in PCOS can completely avoid OHSS and may lead to improved clinical outcomes. We summarized the 10-year work and modified the scheme, including the timely switch from conventional IVF to IVM when patients undergoing COS exhibited features indicating a high risk of unexpected poor ovarian response (UPOR). This study aimed to compare the clinical outcomes in evaluating the efficiency of switching IVF to IVM in UPOR with PCOS and investigate the potential influence of these factors, including general characteristics of the couples, laboratory indicators and variables during the treatment of natural IVM and switching from IVF to IVM cycles, on the risk for clinical pregnancy results. Our data reinforce knowledge in this field, which currently has a limited number of studies.

## 2. Materials and Methods

This retrospective cohort study enrolled 531 women who underwent 588 cycles between 2008 and 2017 at the Center for Reproductive Medicine, Peking University Third Hospital. The study was approved by the Ethics Committee of the Peking University Third Hospital. All participants provided informed consent for the procedures, and all information was handled confidentially.

### 2.1. Study Design

Among the 588 cycles (natural IVM cycles, 377; switching IVF to IVM cycles, 211), 491 cycles underwent fresh embryo transfer (ET), whereas 101 cycles underwent frozen embryo transfers (Figure 1).

Infertile women diagnosed with PCOS according to the revised Rotterdam criteria (Rotterdam ESHRE/ASRM-Sponsored PCOS Consensus Workshop Group, 2004) [19] scheduled for their IVF attempt were eligible to participate in the IVM. The inclusion criteria for the switching IVF to IVM group were designed to incorporate PCOS women whose characteristics indicated a high risk of developing unexpected poor ovarian responses (UPOR). For instance, in the COS process, UPOR is defined as follicular developmental retardation (only one to two mature follicles ≥12 mm or less) and more than 20 follicles (≤10 mm) after being subjected to COH for 8–10 days. The exclusion criteria were fertility preservation and preimplantation genetic testing cycles. The primary outcome measure was the cumulative live birth rate (cLBR), and the secondary outcomes included laboratory and clinical outcomes, maternal safety and obstetric and perinatal complications.

### 2.2. Natural In Vitro Maturation Cycle

Among the 377 natural IVM cycles, no Gn or human chorionic gonadotropin (hCG) was used in the entire process. Participants visited the outpatient on days 6–8 of the menstrual cycle for a transvaginal ultrasound examination to exclude the development of a dominant follicle. Oocyte retrieval was scheduled once the endometrial thickness reached at least 6 mm and the largest follicle was less than 10 mm.

### 2.3. Controlled Ovarian Stimulation (COS) IVF and Switching to IVM Cycle

Among the 211 cycles, the COS protocol for each patient is based on the body mass index (BMI) and previous ovarian response on days 2–4 of the menstrual cycle at the discretion of the patient’s or physician’s preference. There are four protocols, including the GnRH—an ultra-long protocol, GnRH—a long protocol during the mid-luteal phase of their pre-stimulation cycle, GnRH antagonist protocol, and minimal stimulation protocol.

Oocyte retrieval was performed transvaginally 34–36 h after r-hCG (Ovitrelle, Merck Serono, Geneva, Switzerland) injection, and mature oocytes were retrieved from both the leading and small follicles.

### 2.4. Collection, Culture and Maturation of Oocytes

Transvaginal ultrasound-guided oocyte retrieval was conducted with a single-lumen 19G aspiration needle (K-OPS-7035-REH-ET, Cook, Queensland, Australia) under 90 mmHg suction pressure using intravenous sedation. Oocytes were examined on the day of aspiration for maturity after the denudation of cumulus cells. Mature oocytes were inseminated by conventional IVF or intracytoplasmic sperm injection (ICSI) on the day of oocyte retrieval. Immature oocytes were transferred into IVM medium, after 28–32 h of culture, and the oocyte maturation process was evaluated. All metaphase II (MII) oocytes were inseminated using ICSI.

### 2.5. Embryo Morphology Assessment and ET

The cleavage of embryonic development was assessed according to the developmental stage and degree of cytoplasmic fragmentation according to the Istanbul consensus [20]. Additionally, blastocysts were subjected to morphological evaluation according to the Gardner and Schoolcraft grading system [21]. Fresh ET or elective freezing of all embryos was decided by physicians according to the conditions of the women. Excess cleavage stage embryos were vitrified or further cultured to the blastocyst stage, whereas excess blastocysts were vitrified accordingly.

### 2.6. Luteal Phase Supplementation

Patients started oral estradiol valerate (EV) (Primogyn, Bayer, Mexico City, Mexico) on the day of oocyte retrieval and received 6–8 mg daily. Luteal support was provided with oral dydrogesterone (Duphaston, Abbott, Chicago, IL, USA) at 20 mg twice daily and vaginal progesterone gel (Crinone, Merck Serono, Watford, UK) at 90 mg per day starting on the day of oocyte retrieval and continued along with EV until 12 weeks of gestation if the patient was pregnant.

### 2.7. Variables and Outcome Measures

Demographic and clinical characteristics of eligible patients were collected, including age (classified as <35 and ≥35 years), BMI (kg/m^2^), duration of the attempt to conceive, previous conception and indications for IVF.

Laboratory and treatment measures included follicle-stimulating hormone (FSH), luteinizing hormone (LH), E2 on baseline and hCG trigger day, progesterone (P) level on hCG trigger day, androgen level, duration of the follicular phase and total Gn dose.

For fertilization, the number of oocytes retrieved, two pronuclear zygotes, available embryos, good quality embryos and embryos transferred for all patients, as well as differences between the natural IVM and switching IVF/M groups, were observed.

Pregnancy outcomes included a positive β-hCG test (serum hCG level > 5 mIU/mL), clinical pregnancy and live birth (delivery of one or more living infants ≥ 28 weeks’ gestation or birth weight > 1000 g). In cases of successful delivery, we reported gestational age at delivery for preterm labor or term labor and mode of delivery (cesarean section or natural delivery). The safety outcomes included OHSS, miscarriage, ectopic pregnancy and obstetric and perinatal complications.

### 2.8. Statistical Analysis

All analyses were performed using the Statistical Package for Social Sciences version 26.0 (released in 2019, IBM corp., Armonk, NY, USA). A two-sided test was used, and statistical significance was set at *p* < 0.05. For continuous variables, if the parameters were normally distributed, they were presented as mean with standard deviation and compared using Student’s *t*-test; if the parameters were non-normally distributed, their medians and inter-quantile ranges were reported, and the Mann–Whitney U test was utilized to test the distribution of these variables as well. For categorical variables, we presented the proportion between each group, and distributions were compared using Pearson’s chi-square test and Fisher’s exact test when appropriate. Multiple regression analysis was used to analyze the factors that may affect the live birth rate after natural IVM or switching to IVF/M. Results from the logistic regression were then presented with odds ratios (ORs) and their confidence intervals (95% CIs) around the point estimate.

## 3. Results

### 3.1. Participants

In total, 531 women underwent 588 natural IVM or switched to IVF/M cycles (Table 1). Of these cycles, 34 patients underwent two cycles, three patients underwent three cycles, three patients underwent four cycles and two patients underwent five cycles. The number of cycles per year was 2014 (n = 89, 15.1%), 2011 (n = 84, 14.3%), 2009 (n = 76, 12.9%) and 2013 (n = 72, 12.2%), and 25.7% of these cycles were for primary infertility (n = 151). Among the combined causes of infertility, the male sex accounted for 36% (n = 211), followed by tubal disorder (n = 160, 27.2%), previous POR (n = 16, 2.7%), and endometriosis (n = 4, 0.7%). Among the different types of IVF cycles, approximately 64.1% were natural cycles (n = 377), 33.7% were hyperstimulation cycles (n = 198) and the remaining 2.2% were mini-stimulation cycles (n = 13).

### 3.2. Demographic and Clinical Characteristics

The demographic and clinical characteristics of the 588 cycles (377 natural IVM cycles and 211 switching IVF/M cycles) are shown in Table 2. Overall, the mean age of the women was 30.1 ± 3.9 years. The mean age of women in the IVM group was higher than that in the IVF/M group (30.4 ± 3.9 vs. 29.6 ± 3.8 years, *p* = 0.01), while the median BMI showed the opposite distribution (for women with a median BMI of 24.8 kg/m^2^ in the IVM group vs. 26.2 kg/m^2^ in the IVF/M group, *p* < 0.01). Patients in the natural IVM group tended to have a higher antral follicle count (AFC) in both ovaries than those in the switching IVF/M group (both *p* < 0.01). There were no significant findings in the baseline hormone tests of the study participants.

### 3.3. Treatment and Laboratory Measures

During the stimulation treatment, patients in switching IVF/M cycles received a median of 9 days for the follicular phase, with a median dose of 1200 IU Gn (Table 3). However, patients in the natural IVM cycle did not receive Gn for ovarian stimulation. Thus, in the IVF/M cycle, a higher level of E2 and lower level of LH were noted compared with those in the IVM cycle (749.0 pmol/L in IVF/M cycle vs. 220.0 pmol/L in IVM cycle). The LH level was 1.3 mIU/mL in the IVF/M cycle and 8.4 mIU/mL in the IVM cycle. We obtained a median of 13.5 oocytes and 12.0 oocytes (*p* < 0.01) in the natural IVM and switching IVF/M groups, respectively. The number of good quality embryos was 2.2 ± 2.5 in IVM vs. 2.1 ± 2.3 in IVF/M group (*p* = 0.64). Differences between the two groups regarding the number of 2 pronuclear zygotes, number of available embryos and number of embryos transferred were not significant.

### 3.4. Pregnancy and Neonatal Outcomes

The findings on pregnancy and neonatal outcomes in terms of pregnancy, abortion and live birth rates are described in Table 4. A total of 491 cycles of fresh ET were noted among the 588 initial cycles (309 transfer cycles in natural IVM and 182 transfer cycles in switching IVF/M). A total of 161 clinical pregnancies (32.8%) were obtained among the total transfer cycles, with clinical pregnancy rates of 36.6% (n = 113) in the IVM cycles and 26.4% (n = 48) in the IVF/M cycles (*p* = 0.02). The live birth rate (LBR) in the natural IVM group (24.3%) was higher than that in the IVF/M group (16.4%) (*p* = 0.04). The cancelation of the treatment cycles was 97 cycles without ET treatment, cycles canceled due to nine cases of poor-quality eggs, 16 cases of endometrial factors, four cases of failure to retrieve any eggs, 65 cases of no embryos available for transfer and three cases of preventing the incidence of OHSS.

The cumulative results for one complete cycle are listed in Table 4. There were 592 cycles of fresh and frozen ET (369 transfer cycles in natural IVM and 223 transfer cycles in switching IVF/M). Among the total transfer cycles, 191 clinical pregnancies (32.3%) were obtained, with clinical pregnancy rates of 36.0% (n = 133) for IVM cycles and 26.0% (n = 58) for IVF/M cycles (*p* = 0.01). In women undergoing the complete cycles, the cumulative live birth rate (cLBR) within a complete cycle was 23.6% in natural IVM and 17.4% in switching IVF/M, respectively (*p* = 0.05).

Three fetal malformations (two cases during the prenatal diagnosis of fetal cardiac malformations and one of the vascular abnormalities of the umbilical cord) were found during pregnancy, and all these women received induced labor, which was counted as a miscarriage. Finally, 115 deliveries (19.8%) were performed, with 124 healthy babies (99.0%) and one stillbirth (1.0%). There were 102 singletons (98.1%) and 22 twins (17.7%) among live births, and the sex distribution among all deliveries was 65 males (52.0%) and 60 females (48%). Among all 115 deliveries, 18.9% (n = 21) were preterm labor and 81.1% (n = 90) were cesarean section, with mean birth weights of 3155.6 ± 723.0 for singleton and 2707.3 ± 521.2, respectively.

### 3.5. Factors Associated with Live Birth after IVM

In a univariate analysis of factors associated with live birth after natural IVM or switching IVF/M, the only significant factor was the number of embryos transferred (Table 5). Transferring two embryos could increase live birth by approximately 90% when compared with transferring only one embryo (*p* < 0.01), while transferring three embryos could not significantly increase the relevant LBR (*p* = 0.36). After the stepwise logistic analysis, a number of factors were entered into the multivariable logistic regression (including androgen level, duration of the follicular phase, number of oocytes retrieved and number of embryos transferred). However, the only significant independent factor associated with live birth after IVM on multivariate logistic regression analysis was the number of embryos transferred (OR = 0.07, 95% CI: 0.01–0.59, *p* = 0.02).

## 4. Discussion

The present study focused on the management of UPOR in patients with PCOS and showed that switching to IVF/M could be a feasible choice in PCOS patients who failed to achieve adequate coordination of multiple follicular growths due to an under-response to COS. Our retrospective data summarized the natural IVM and switching to IVF/M with a rescue strategy that had been used in our clinic from 2008 to 2017.

IVF with COS is a highly effective treatment for women with PCOS; however, PCOS can have contrasting responses to COS, and ovarian responses to the same COS protocols may vary considerably among different PCOS patients and even among different cycles in the same patient [15,22]. The management of women with PCOS and UPOR can be an equally frustrating challenge. Currently, these patients are presented with the following choices: (1) canceling the cycle, (2) conversion to intrauterine insemination (IUI), or (3) continuing with Gn stimulation regardless of the poor response and finishing the oocyte retrieval and transfer of embryos when available [23]. In this study, UPOR referred specifically to patients with a normal ovarian reserve and ample AFC, yet showed a poor hormonal and follicular response in COS. The decision to proceed with the COS cycle is considered, and it is unclear whether IVF or conversion to IVM is the best choice.

The AFC (12–20 or >20) in the IVF/M group was significantly less than that in the IVM group; therefore, these patients were initially more suitable to use the conventional COS treatment [16,24,25]. It is unclear why women with PCOS show contrasting responses to Gn stimulation. Patients had no or only a few (<3) dominant follicles with low E2 levels in the COS process; however, the high number of small follicles per ovary (2–3 times that of normal) leads to poor follicular growth, which adversely affects mature oocyte retrieval, and more importantly, reduces the number of viable oocytes and embryos and the probability of conception. We changed the treatment strategies to switch IVF treatment to immature egg retrieval followed by IVM of immature oocytes. In the current report, women in the IVF/M group received an average number of 12.0 oocytes retrieved, and the clinical pregnancy rate was 26.4%. Interestingly, of the 182 fresh ET cycles in the IVF/M switching, none developed OHSS. For UPOR females switching to IVM, the possibility of OHSS is very low. If UPOR high-risk PCOS females continue Gn administration, the incidence of OHSS may significantly increase. Previous studies reported that in patients who developed a severe form of the syndrome, 95% of pre-ovulatory follicles were <16 mm and most were <9 mm. This suggests that the small and medium (<14 mm) size follicles are mostly responsible for the high serum estradiol concentrations (and vasoactive compounds) [26]. Hence, the timely switch to IVM treatment can prevent OHSS, while ensuring access to fresh ET, thereby reducing the cycle cancelation rate.

In our previous data, the LBR was 8.8–9.1% after IUI with the husband’s or partner’s semen [27], but the live birth rate of switching IVF/M cycles was 16.4%, and the cLBR was 17.4%. Our results provide a good example of UPOR to exogenous Gn and the application of IVM in PCOS. Another objective fact is that most reproductive centers have adopted criteria for cancelation of stimulation cycles that do not produce an adequate number of mature follicles, which in many centers is a minimum of four [28,29,30,31]. At our center, in the IVF/M cycle, treatment is continued, regardless of the number of developing follicles. IVF attempts are the only treatment option in patients who are <35 years with a severe male factor or tubal factor infertility.

Previous studies have shown that IVM is a successful and widely practiced method and a feasible alternative to conventional IVF in women with a high AFC [32]. The advantages of IVM over IVF were reported in our recent randomized controlled trial (RCT) [10,33], including the absence of COS, improved convenience, a lower monitoring burden, and the establishment of OHSS-free strategies. Initially, we believed that most follicles normally degenerate to become atretic follicles, which would reach dominance, and the dominant follicle would suppress the development of other follicles and atresia. However, studies have indicated that the developmental competence of immature oocytes derived from small antral follicles is not adversely affected by the dominant follicle [34]. We describe our IVM procedure modified in 2008, successful pregnancies and subsequent live births in patients who underwent conventional IVF followed by IVM treatment. Our results indicate that mature and immature oocyte retrieval followed by IVF combined with IVM may be a potential treatment for women with various causes of infertility.

Compared to the results in our recent IVM RCT performed between March 2018 and July 2019, which included 175 PCOS women who underwent the natural IVM protocol, the results of the natural IVM group in this retrospective study showed that the number of oocytes retrieved was 13.5 in the retrospective IVM versus 14 in the RCT IVM, the number of good quality embryos was 2.0 versus 2.0 and the clinical pregnancy rate was 63.2% in retrospective IVM versus 38.5% in RCT IVM (36.6% vs. 32.5%), and the live birth rates (24.3% vs. 22.5%) [33]. The summary data suggest the potential for IVM in our center to be successful during the past 15 years. IVM may be an attractive method for many patients with many advantages, and we believe that IVM could be a good alternative not only for PCOS but also for women with different etiologies when the conditions are optimized, including for fertility preservation, resistant ovary syndrome, poor responders, eliminating the risk of OHSS in PCOS and for women with oocyte maturation defects.

However, several limitations associated with this study warrant mention. Due to the study’s retrospective nature and the availability of linked data between 2008 and 2017, some potential confounding factors are not adjusted. Furthermore, ART methodology and efficacy have significantly changed over the 9-year recruitment to the study and therefore, this may affect the result and conclusions drawn. Nevertheless, it should be noted that the follow-up evaluation was conducted via phone calls in our study, and it is unclear how reliable or accurate these phone call reports are. Further studies should be carried out among women with PCOS with different phenotypes to segregate the role of hyperandrogenism and metabolic derangements, and clinical evaluation and metabolic parameters including those of AMH, androgens, triglycerides, high-density lipoproteins and glucose tolerance.

In conclusion, timely switching IVF/IVM for patients with PCOS in UPOR tendency provides a feasible clinical approach for infertility treatment. The results should be interpreted with caution owing to the retrospective design, and larger studies are needed to validate them.

## Figures and Tables

**Figure 1 jcm-12-01978-f001:**
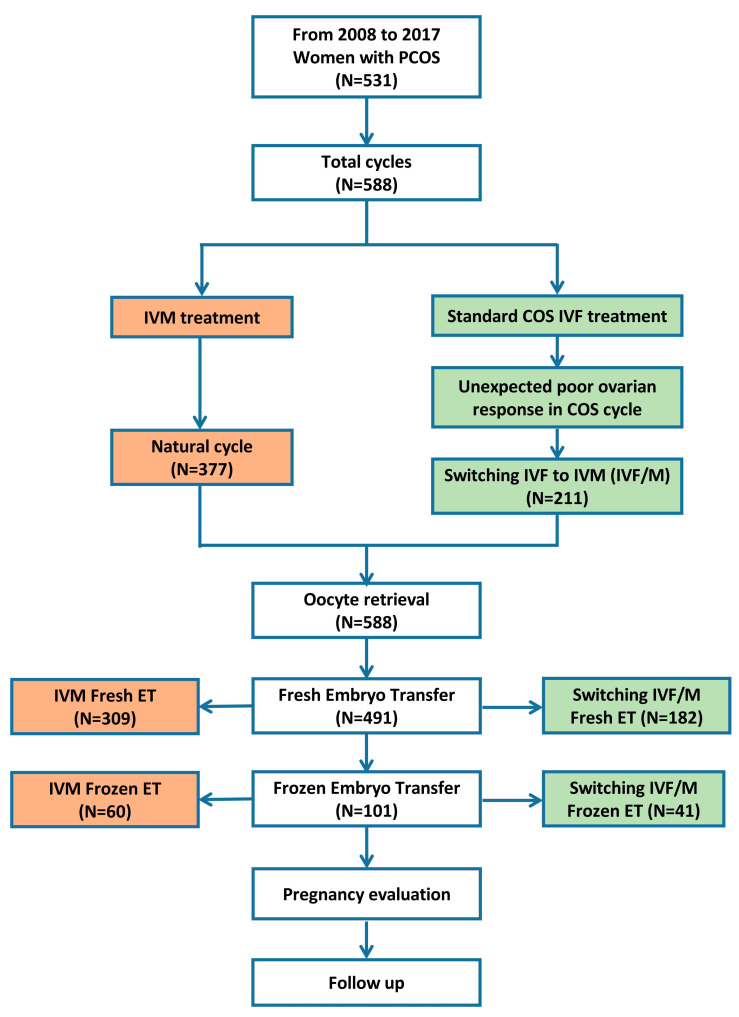
Flowchart of this retrospective cohort study. COS, controlled ovarian stimulation; IVF, in vitro fertilization; IVM, in vitro maturation.

**Table 1 jcm-12-01978-t001:** General information of database.

	No. of Cycles	Percentages (%)
Total numbers	588	100.00
Year		
2008	29	4.9
2009	76	12.9
2010	54	9.2
2011	84	14.3
2012	43	7.3
2013	72	12.2
2014	89	15.1
2015	52	8.8
2016	56	9.5
2017	33	5.6
Type of infertility		
Primary infertility	151	25.7
Secondary infertility	437	74.3
PCOS for single cause for infertility	197	33.5
PCOS combined causes for infertility		
Male factors	211	35.9
Tubal disorder	160	27.2
Previous poor ovarian response	16	2.7
Endometriosis	4	0.7
Type of ART cycle		
Natural cycle	377	64.1
Hyper-stimulation cycle	198	33.7
GnRH-a protocol	128	64.6
Short GnRH—a protocol	22	17.2
Long GnRH—a protocol	78	60.9
Super long GnRH-a protocol	28	21.9
GnRH-ant protocol	70	35.4
Mini-stimulation cycle	13	2.2

Abbreviation: PCOS, polycystic ovary syndrome; ART, assisted reproductive technology; GnRH-a, Gonadotrophin-releasing hormone agonist; GnRH-ant, Gonadotrophin-releasing hormone antagonist.

**Table 2 jcm-12-01978-t002:** Demographic and clinical characteristics of patients, by the natural or stimulation cycle.

	Total (n = 588)	Type of Cycle		*p*-Value
		IVF/M Cycle (n = 211)	IVM Cycle (n = 377)	
Age				
Mean (SD)—years	30.1 (3.9)	29.6 (3.8)	30.4 (3.9)	0.01
Distribution—no. (%)				
<35 years	503 (85.5)	185 (87.7)	318 (84.4)	0.27
≥35 years	85 (14.5)	26 (12.3)	59 (15.6)	
BMI ^a^ (median, IQR)—kg/m^2^	25.3 (22.4–28.4)	26.2 (23.1–28.9)	24.8 (22.3–28.2)	0.01
Fertility history				
Duration of attempt to conceive (median, IQR)—year	4.0 (2.0–6.0)	4.0 (2.0–6.0)	4.0 (3.0–6.0)	0.15
Previous conception—no. (%)	151 (25.7)	53 (25.1)	98 (26.0)	0.45
Indications for IVF—no. (%)				
Male factors	211 (35.9)	91 (43.1)	120 (31.8)	0.01
Tubal disorder	160 (27.2)	68 (32.2)	92 (24.4)	0.04
Previous poor ovarian response	16 (2.7)	4 (1.9)	12 (3.2)	0.36
Endometriosis	4 (0.7)	0 (0.0)	4 (1.1)	0.13
Number of IVF attempt				
Median (IQR)	1.0 (1.0–2.0)	1.0 (1.0–1.0)	1.0 (1.0–2.0)	0.99
Distribution-no. (%)				
1	423 (71.9)	177 (83.9)	246 (65.3)	<0.01
2	94 (16.0)	21 (10.0)	73 (19.4)	
3	35 (6.0)	8 (3.8)	27 (7.2)	
4	21 (3.6)	3 (1.4)	18 (4.8)	
≥5	15 (2.5)	2 (1.0)	13 (3.5)	
Ultrasonographic findings				
AFC in both ovaries—no. (%)				
<12	84 (16.5)	52 (25.1)	32 (10.6)	<0.01
12–20	375 (73.5)	144 (69.6)	231 (76.2)	
>20	51 (10.0)	11 (5.3)	40 (13.2)	
Endometrial thickness ^b^ (Mean, SD)—mm	8.2 (1.7)	8.9 (1.7)	7.7 (1.6)	<0.01
Laboratory tests				
FSH ^c^ (median, IQR)—mIU/mL	5.8 (4.8–7.0)	5.9 (4.9–7.3)	5.6 (4.6–6.8)	0.67
LH ^d^ (median, IQR)—mIU/mL	7.0 (4.2–10.7)	6.7 (3.5–10.3)	7.1 (4.4–11.0)	0.76
Ratio of LH to FSH ^e^ (median, IQR)	1.2 (0.7–1.8)	1.1 (0.6–1.7)	1.3 (0.7–1.9)	0.07
Estradiol ^f^ (median, IQR)—pmol/L	168.5 (122.0–220.0)	154.0 (119.5–218.5)	171.0 (124.0–221.0)	0.48
Androgen ^g^ (median, IQR)—pmol/liter	11.8 (8.1–17.2)	11.2 (7.8–15.8)	12.1 (8.1–18.9)	0.27

Abbreviations: SD, standard deviation; IQR, interquartile range; BMI, body mass index; PCOS, polycystic ovary syndrome; POR, poor ovarian response; IVF, in vitro fertilization; AFC: antral follicle count. ^a^. Body mass index is the weight in kilograms divided by the square of the height in meters. ^b^. Data regarding endometrial thickness were missing for 107 cycles (18.2%). ^c^. Data regarding FSH levels were missing for 45 cycles (7.7%). ^d^. Data regarding LH were missing for 41 cycles (7.0%). ^e^. LH and FSH levels were measured in units per liter. Data were missing for 48 cycles (8.2%). ^f^. Data regarding estradiol were missing for 40 cycles (6.8%). ^g^. Data regarding androgen were missing for 190 cycles (32.3%).

**Table 3 jcm-12-01978-t003:** Treatment and laboratory outcomes of participants, by the natural or stimulation cycle.

	Total (n = 588)	Type of Cycle		*p*-Value
		IVF/M Cycle (n = 211)	IVM Cycle (n = 377)	
Duration of follicular phase ^a^ (median, IQR)—days	0.0 (0.0–7.0)	9.0 (6.0–12.0)	0.0 (0.0–0.0)	<0.01
Total gonadotropin dose ^b^ (median, IQR)—IU	0.0 (0.0–900.0)	1200.0 (750.0–1950.0)	0.0 (0.0–0.0)	<0.01
Estradiol level on hCG trigger day ^c^ (median, IQR)—pmol/L	374.0 (190.0–852.5)	749.0 (355.0–2061.8)	220.0 (164.0–418.0)	<0.01
Progesterone level on hCG trigger day ^d^ (median, IQR)—nmol/L	0.9 (0.6–1.4)	0.9 (0.6–1.4)	0.9 (0.6–1.4)	0.62
LH level on hCG trigger day ^e^ (median, IQR)—mIU/mL	5.5 (1.3–9.4)	1.3 (0.4–3.6)	8.4 (5.9–13.0)	<0.01
No. of oocytes retrieved ^f^				
Total number	8779	2786	5993	<0.01
Median (IQR)	13.0 (8.0–20.0)	12.0 (7.0–17.0)	13.5 (8.0–21.0)	
Method of fertilization—no. (%)				
Conventional IVF	11 (1.9)	8 (3.8)	3 (0.8)	0.01
ICSI	570 (96.9)	198 (93.8)	372 (98.7)	
Half-ICSI	7 (1.2)	5 (2.4)	2 (0.5)	
No. of mature oocytes				
Total number	4184	1573	2611	0.40
Median (IQR)	6.0 (4.0–10.0)	6.0 (4.0–10.0)	6.0 (4.0–10.0)	
No. of 2 pronuclear zygotes				
Total number	2580	957	1623	0.45
Median (IQR)	4.0 (2.0–6.0)	4.0 (2.0–6.0)	3.0 (1.0–6.0)	
No. of available embryos				
Total number	1515	522	993	0.46
Median (IQR)	2.0 (1.0–3.0)	2.0 (2.0–3.0)	2.0 (1.0–3.0)	
No. of good quality embryos				
Total number	1241	432	809	0.64
Median (IQR)	2.0 (0.0–3.0)	2.0 (0.0–3.0)	2.0 (0.0–3.0)	
Patients undergoing the first fresh embryo transfer—no. (%)				
Day 3	485 (98.8%)	179 (98.4%)	306 (99.0%)	0.5
Day 5	6 (1.2%)	3 (1.6%)	3(0.1%)	
No. of embryo transferred (the first Fresh ET cycle)				
Total number	975	356	619	
Mean (SD)	1.7 (0.9)	1.7 (0.8)	1.6 (0.9)	0.53
Distribution—no. (%)				
1	71 (12.1)	24 (11.4)	47 (12.5)	0.06
2	356 (60.5)	142 (67.3)	214 (56.8)	
3	64 (10.9)	16 (7.6)	48 (12.7)	
No. of embryo transferred (the cumulative ET after one complete cycle)				
Total number	1167	432	735	
Mean (SD)	2.0 (0.5)	2.0 (0.6)	1.9 (0.3)	0.73
Distribution—no. (%)				
1	94 (16.0)	36 (17.1)	58 (15.4)	0.188
2	411 (70.0)	163 (77.3)	249 (66.0)	
3	83 (14.1)	24 (11.4)	59 (15.7)	

Abbreviations: SD, standard deviation; IQR, interquartile range; IVF, in vitro fertilization; ICSI, intracytoplasmic sperm injection. ^a^ Data regarding the duration of the follicular phase were missing for one cycle (0.2%). ^b^ Data regarding the total gonadotropin dose were missing for six cycles (1.0%). ^c^ Data regarding estradiol levels on hCG trigger days were missing for 139 cycles (23.6%). ^d^ Data regarding progesterone levels on hCG trigger days were missing for 143 cycles (24.3%). ^e^ Data regarding LH levels on hCG trigger days were missing for 147 cycles (25.0%). ^f^ Data regarding the number of oocytes retrieved missing for three cycles (0.5%).

**Table 4 jcm-12-01978-t004:** Pregnancy and neonatal outcomes of participants, by the natural or stimulation cycle.

	Total (n = 588)	Type of Cycle		*p*-Value
		IVF/M Cycle (n = 211)	IVM Cycle (n = 377)	
Number of transferred cycles—(the first fresh ET cycle)	491	182	309	
Number of embryos transferred	975	356	619	
Positive β-hCG—no./total no. (%)	198/491 (40.3)	63/182 (34.6)	135/309 (43.7)	0.05
Clinical pregnancy-no./total no. (%)	161/491 (32.8)	48/182 (26.4)	113/309 (36.6)	0.02
Ectopic pregnancy	7/161 (4.3)	3/48 (6.3)	4/113 (3.5)	0.44
Uterine pregnancy	154/161 (95.7)	45/48 (93.8)	109/113 (96.5)	0.14
Number of ET embryos				0.49
1	37/154 (24.0)	12/45 (26.7)	25/109 (22.9)	
2	114/154 (74.0)	33/45 (73.3)	81/109 (74.3)	
3	3/154 (1.9)	0/45 (0.0)	3/109 (2.8)	
Implantation (per embryo)—no./total no. (%)	161/975 (16.5)	48/356 (13.5)	113/619 (18.3)	0.05
Pregnancy loss—no/total no. (%)				
Among biochemical pregnancy	37/198 (18.7)	15/63 (23.8)	22/135 (16.3)	0.21
Among clinical pregnancy	48/161 (29.8)	14/48 (29.2)	34/113 (30.1)	0.91
First trimester	37/161 (23.0)	11/48 (22.9)	26/113 (23.0)	
Second trimester	11/161 (6.8)	3/48 (6.3)	8/113 (7.1)	
Live birth rate(after 28 weeks gestation)-no./total no. (%)	105/491 (21.4)	30/182 (16.4)	75/309 (24.3)	0.04
Number of transferred cycles—(the cumulative ET after one complete cycle)	592	223	369	
Number of embryos transferred	1167	432	735	
Positive β-hCG—no./total no. (%)	231/592 (39.0)	73/223 (32.7)	158/369 (42.8)	0.01
Clinical pregnancy-no./total no. (%)	191/592 (32.3)	58/223 (26.0)	133/369 (36.0)	0.01
Ectopic pregnancy	8/191 (4.2)	3/58 (5.2)	5/133 (3.8)	0.65
Uterine pregnancy	185/191 (96.9)	55/58 (94.8)	130/133 (97.7)	0.29
Number of ET embryos				
1	51/185 (27.6)	14/55 (25.5)	37/130 (28.5)	0.89
2	126/185 (79.7)	39/55 (70.9)	87/130 (66.9)	
3	8/185 (4.3)	2/55 (3.6)	6/130 (4.6)	
Implantation (per embryo)—no./total no. (%)	191/1167(16.4)	58/ 432(13.4)	133/735 (18.1)	0.03
Pregnancy loss—no/total no. (%)				
Among biochemical pregnancy	40/231 (17.3)	15/73 (20.5)	25/158 (15.8)	0.38
Among clinical pregnancy	59/191 (30.9)	16/58 (27.6)	43/133 (32.3)	0.51
First trimester	47/191 (24.6)	13/58 (22.4)	34/133 (25.6)	
Second trimester	12/191 (6.3)	3/58 (5.2)	9/133 (6.9)	
Live birth rate (after 28 weeks of gestation)-no./total no. (%)	125/592 (21.1)	38/223 (17.4)	87/369 (23.6)	0.05
Pregnancy complication-no./total no. (%)				
Hypertensive disorders	3/185 (1.62)	0/55 (0.0)	3/130 (2.31)	0.26
Premature rupture of membranes	2/85 (2.35)	0/55 (0.0)	2/130 (1.54)	0.36
Gestational diabetes mellitus	1/185 (0.54)	0/55 (0.0)	1/130 (0.77)	0.51
Anemia	1/185 (0.54)	0/55 (0.0)	1/130 (0.77)	0.51
Intrauterine infection	1/185 (0.54)	1/55 (1.82)	0/130 (0.0)	0.12
Delivery ^a^—no./total no. (%)	115/582 (19.8)	35/220(15.9)	80/362 (22.1)	0.07
Number of neonates ^b^	125	38	87	
Live birth	124/125 (99.0)	37/38 (97.4)	87/87 (100.0)	0.13
Singleton	102/124 (98.1)	31/37 (83.8)	69/87 (79.3)	0.56
Twin	22/124 (17.7)	6/37 (16.2)	18/87 (20.7)	0.56
Stillbirth	1/125 (1.0)	1/38 (2.6)	0/87 (0.0)	0.19
Sex				
Male	65/125 (52.0)	20/38 (52.6)	45/87 (51.7)	0.92
Female	60/125 (48.0)	18/38 (47.4)	42/87 (48.3)	
Gestational age on delivery (weeks)				
Medianc (IQR)—weeks	38.0 (37.0–39.0)	38.0 (35.8–39.0)	38.0 (37.0–39.0)	0.99
Distribution-no./total no. (%) ^c^				
Preterm labor (28–36 + 6 weeks)	21/111(18.9)	5/35 (14.3)	16/76 (21.1)	0.39
Term labor (≥37 weeks)	90/111 (81.1)	30/35 (85.7)	60/76 (78.9)	0.40
Mode of delivery-no./total no. (%)				
Cesarean section	89/114 (78.1)	26/34 (76.5)	63/80 (78.8)	0.78
Natural delivery	25/114 (21.9)	8/34 (23.5)	17/80 (21.2)	0.57
Advanced neonatal outcome-no./total no. (%)				
Birth defect	3/185 (1.6)	0/55 (0.0)	3/130 (2.3)	0.26
Neonatal asphyxia	0/185 (0.0)	0/55 (0.0)	0/130 (0.0)	
Neonatal death	0/185 (0.0)	0/55 (0.0)	0/130 (0.0)	
Birth weight—g				
Singleton ^d^	3155.6 (723.0)	3089.6 (800.5)	3242.2 (688.3)	0.78
Twin	2707.3 (521.2)	2701.3 (534.9)	3112.3 (593.1)	0.81

Abbreviations: β-hCG, β-human chorionic gonadotropin; SD, standard deviation; IQR, interquartile range. ^a^ Data regarding delivery were missing for 10 cycles (2.0%). ^b^ Data regarding the number of neonates were missing for two cycles (0.4%). ^c^ Data regarding gestational age at delivery on hCG trigger day were missing for four cycles (0.8%). ^d^ Data regarding birth weight were missing for one cycle (1.0%).

**Table 5 jcm-12-01978-t005:** Factors associated with the live birth rate after the first embryo transfer of IVM or IVF/M.

	Live Birth (N = 104)	No Live Birth (N = 377)	Unadjusted Model		Adjusted Model A	
			Odds Ratio (95% CI)	*p*-Value	Odds Ratio (95% CI)	*p*-Value
Age—year	30.4 (3.7)	29.8 (3.8)	1.04 (0.98–1.10)	0.23	-	-
BMI ^a^—kg/m^2^	25.3 (4.1)	25.8 (4.2)	0.97 (0.92–1.03)	0.28	-	-
Duration of attempt to conceive—year	4.6 (2.8)	4.6 (2.9)	1.00 (0.93–1.08)	0.98	-	-
Previous conception—no. (%)	28 (29.5)	99 (25.0)			-	-
Indications for PCOS IVF—no. (%)					-	-
Tubal disorder	27 (28.4)	106 (26.8)	0.92 (0.56–1.52)	0.75	-	-
POR	0 (0.0)	9 (2.3)	-	-	-	-
Male factors	35 (36.8)	141 (35.6)	0.95 (0.50–1.51)	0.82	-	-
Endometriosis	0 (0.0)	2 (0.5)	-	-	-	-
Type of stimulation protocol—no. (%)					-	-
IVM	68 (71.6)	241 (60.9)	0.62 (0.38–1.01)	0.05	-	-
IVF/M	27 (28.4)	155 (39.1)	Ref	ref	-	-
Number of IVF attempt	1.4 (0.7)	1.5 (1.1)	0.90 (0.71–1.15)	0.41	-	-
AFC in both ovaries	30.0 (11.5)	29.4 (12.5)	1.00 (0.98–1.02)	0.89	-	-
Endometrial thickness ^b^—mm	8.2 (1.8)	8.3 (1.6)	0.98 (0.85–1.13)	0.73	-	-
FSH ^c^—mIU/mL	6.2 (2.3)	6.7 (7.8)	0.99 (0.95–1.03)	0.60	-	-
LH ^d^—mIU/mL	8.4 (5.0)	8.2 (6.3)	1.01 (0.97–1.04)	0.77	-	-
Estradiol ^e^—pmol/L	171.5 (75.8)	181.6 (121.4)	1.00 (0.99–1.00)	0.45	-	-
Androgen ^f^—pmol/liter	14.4 (14.5)	13.3 (7.0)	1.00 (0.99–1.04)	0.37	1.02 (0.99–1.05)	0.24
Duration of follicular phase ^g^—days	2.6 (4.7)	3.6 (5.2)	0.96 (0.92–1.01)	0.12	0.94 (0.88–1.01)	0.07
Total gonadotropin dose ^h^ (SD)—IU	389.8 (775.4)	538.1 (875.9)	1.00 (0.99–1.00)	0.14	-	-
Estradiol level on hCG trigger day ^i^ (SD)—pmol/L	948.9 (2077.7)	1406.9 (3438.5)	1.00 (1.00–1.00)	0.29	-	-
Progesterone level on hCG trigger day ^j^ (SD)—nmol/L	1.1 (0.7)	1.2 (1.0)	0.91 (0.67–1.23)	0.54	-	-
LH level on hCG trigger day ^k^—mIU/mL	11.2 (24.4)	7/0 (12.4)	1.01 (1.00–1.03)	0.90	-	-
Number of oocytes retrieved ^l^ (SD)	16.3 (10.0)	16.1 (10.5)	1.00 (0.98–1.02)	0.89	0.96 (0.92–1.00)	0.05
Number of embryos transferred-no./total no. (%)						
1	2 (2.1)	69 (17.4)	ref	ref	ref	ref
2	76 (80.0)	280 (70.7)	0.08 (0.02–0.36)	<0.01	0.07 (0.01–0.59)	0.02
3	17 (17.9)	47 (11.9)	0.75 (0.41–1.38)	0.36	0.60 (0.24–1.49)	0.27

Abbreviations: SD, standard deviation; IQR, interquartile range; BMI, body mass index; PCOS, polycystic ovary syndrome; POR, poor ovarian response; IVF, in vitro fertilization; AFC: antral follicle count. ^a^ The body-mass index is the weight in kilograms divided by the square of the height in meters. ^b^ Data regarding endometrial thickness were missing for 107 cycles (18.2%). ^c^ Data regarding FSH levels were missing for 45 cycles (7.7%). ^d^ Data regarding luteinizing hormone were missing for 41 cycles (7.0%). ^e^ Data regarding estradiol were missing for 40 cycles (6.8%). ^f^ Data regarding androgen were missing for 190 cycles (32.3%). ^g^ Data regarding the duration of the follicular phase were missing for one cycle (0.2%). ^h^ Data regarding the total gonadotropin dose were missing for six cycles (1.0%). ^i^ Data regarding estradiol levels on hCG trigger days were missing for 139 cycles (23.6%). ^j^ Data regarding progesterone levels on hCG trigger days were missing for 143 cycles (24.3%). ^k^ Data regarding LH levels on hCG trigger days were missing for 147 cycles (25.0%). ^l^ Data regarding the number of oocytes retrieved were missing for three cycles (0.5%).

## Data Availability

The datasets used in the current study are available from the corresponding author upon reasonable request.

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
