# Peer review of "Effectiveness, Flexibility and Safety of Switching IVF to IVM as a Rescue Strategy in Unexpected Poor Ovarian Response for PCOS Infertility Patients"

_jcm, 2023, doi:10.3390/jcm12051978_

Round 1

Reviewer 1 Report

Dear All,

Congratulations on your article and the massive data about IVM that you brought to our attention.

I only suggest some minor revisions:

page 1 raw 2, the use of  "switching IVF/M" in the title is not very clear and may be confusing for someone who didn't read the whole article.

Maybe you could use instead switching "IVF to IVM."

page7 table II -some minor errors in the text"FSHc, Estradiolf"

page8 table III-some minor errors in the text"dayc, dayd,daye, IQR(3)"

page12 table V some minor errors in the text " endometrial thicknessb, follicle-stimulating hormones,androgenf, phaseg, doseh

Author Response

SPECIFIC COMMENTS

Congratulations on your article and the massive data about IVM that you brought to our attention.

I only suggest some minor revisions:

*page 1 raw 2, the use of "switching IVF/M" in the title is not very clear and may be confusing for someone who didn't read the whole article.

Maybe you could use instead switching "IVF to IVM."

Response: We appreciate your great help and the opportunity to revise our manuscript. 

We updated the title information. 

*page7 table II -some minor errors in the text"FSHc, Estradiolf"

*page8 table III-some minor errors in the text"dayc, dayd,daye, IQR(3)"

*page12 table V some minor errors in the text " endometrial thicknessb, follicle-stimulating hormones,androgenf, phaseg, doseh

Response: We have updated tables.

Reviewer 2 Report

it is an excellent paper and i have only 2 small comments 

-the sentence starting on line 58 is difficult to understand

-I found always funny to have the advice on an ethical commitee on cases that were previously treated - in my conutry, it is not necessary 

Author Response

SPECIFIC COMMENTS

it is an excellent paper and i have only 2 small comments

*the sentence starting on line 58 is difficult to understand

Response: We appreciate your great help and the opportunity to revise our manuscript. 

We have revised the text in this section.

It now reads: (Line 70)

Chian et al.[17] first demonstrated the pregnancies and live births that resulted from IVF of mature oocytes retrieved from dominant follicles in a natural cycle combined with IVM of immature oocytes retrieved from small follicles, and indicated that the natural cycle IVF/M had been an attractive choice for infertility treatment for various reasons.

*I found always funny to have the advice on an ethical commitee on cases that were previously treated - in my conutry, it is not necessary.

Response: Thanks for your question.

The research was approved by the Ethics Committee of Peking University Third Hospital. All patients received a full explanation of the switching IVF/M protocol and provided written signed informed consent.

Reviewer 3 Report

The authors investigated the effects of switching from IVF to IVM in the treatment of PCOS patients with an unexpected poor ovarian response (UPOR) tendency. There are some comments for revision of this manuscript.

1.    In the manuscript, the authors have mentioned no OHSS in the patients who underwent IVF/M. As you know, the possibility of OHSS in patients with the poor ovarian response is very low. Therefore, the research on patients with a high ovarian response may reveal an answer to this question. I suggest discussing this issue in the discussion section and revising the title based on your primary outcomes.

2.    Grammatical errors:

- The Sentence “No statistically significant differences in the number of 2 pronuclear (2PN) and available embryos.” (Line 31) has a grammatical error.

-The sentence “Chian et al. [17] first demonstrated that a natural IVF cycle…” (Line 58) has grammatical errors and is not clear. Please rewrite it.

-“The number of good quality embryos was 2.2±2.5 IVM…” (Line 208)

- Please rewrite the sentence “Our data reinforce knowledge in this field, which currently has a limited number of studies on this 71 topic.” (Line 70)

3.    you need to consider the inclusion and exclusion criteria in more detail. If your PCOS patients were with UPOR, you should mention this point as well.  

4.    There is a lot of heterogenicity in the study groups. As the results showed significant differences in age, BMI, method of fertilization, etc. How can you explain the effects of these factors on your results?

5.    Keywords:

Please revise according to alphabetical order.

6.    Abbreviations:

All abbreviations should have been provided in full on the first mention and this applies to the title, abstract, main text, and each table/figure as they will be read independently.

Please explain these words when they appear for the first time:

OHSS in the abstract, AFC in line 290, and RCT in line 315. As you defined cLBR for the first time in line 86, please use this abbreviated form thereafter.

It is important to use uniform terminology in your work. For example, there is inconsistency in the PCOS definition as authors have used “polycystic ovary syndrome” in the abstract and “polycystic ovarian syndrome” in keywords, introduction, etc.

Please use FSH instead of “follicle-stimulating hormone” after explaining it for the first time (especially in Tables).

7.    Tables:

Please redesign the tables as they are difficult to read and follow the contents. I suggest to align your contents in the first column with the left margin and use hanging for the subgroups.

The lower cases (a-g) and also the numbers (e.g. Kg/m2) used in tables should be written as superscripts.

Table I:

It seems that the total percentage of some factors such as, “PCOS combined causes for infertility” does not reach 100%. You have not mentioned any missing data!!

Table IV: As you mentioned about the missing number in the table footnote, please remove them from the Table.

The definition of the β-hCG abbreviation is missing from the footnote.

8.    In line 188, you mentioned “Patients in the natural IVM group tended to have more previous IVF attempts than those in the switching IVF/M group” but your data in Table II does not support this statement. Could you please explain it?

9. The sentence “No significant difference was found between IVM and IVF/M cycles in terms of clinical pregnancy rate…” (Line 246) is not consistent with the results of clinical pregnancy in Table IV!!

Author Response

SPECIFIC COMMENTS

The authors investigated the effects of switching from IVF to IVM in the treatment of PCOS patients with an unexpected poor ovarian response (UPOR) tendency. There are some comments for revision of this manuscript.

Response: We appreciate your great help and the opportunity to revise our manuscript. 

*1. In the manuscript, the authors have mentioned no OHSS in the patients who underwent IVF/M. As you know, the possibility of OHSS in patients with the poor ovarian response is very low. Therefore, the research on patients with a high ovarian response may reveal an answer to this question. I suggest discussing this issue in the discussion section and revising the title based on your primary outcomes.

Response: Thanks for your comments.

We have now addressed this comment in the discussion section.(line 287-296)

  • For UPOR females switching to IVM, the possibility of OHSS is very low.
  • If UPORhigh-risk PCOS-females continue Gn administration, the incidence of OHSS may significantly increase. In previous studies report: patients who developed a severe form of the syndrome, 95% of pre-ovulatory follicles were <16 mm and most were <9 mm (Enskog et al., 1999). This suggests that it is the small and medium (<14 mm) size follicles that are mostly responsible for the high serum oestradiol concentrations (and vasoactive compounds) (Enskog et al., 1999). Papanikolaou proposed that the high count of larger follicles (medium/large follicles ≥ 13 follicles; ≥ 11 mm in diameter) is the threshold of OHSS risk with a sensitivity of 84.9% and specificity of 69.0% (Papanikolaou EG et al.,2006). Hence, the timely switch to IVM treatment can prevent OHSS, while reducing the cycle cancelation rate.
  • It was agreed with Zhao’s reports(Zhao et al., 2004; Zhao t al., 2022), that switching from IVF to IVM for patients with OHSS tendency provides a feasible clinical approach for infertility treatment. Their inclusion criteria were designed to incorporate women whose characteristics indicated a high risk of developing OHSS, for instance, the presence of > 30 follicles in both ovaries and a follicular diameter < 12 mm after being subjected to COH for 5 – 12 days.

*2. Grammatical errors:

- The Sentence “No statistically significant differences in the number of 2 pronuclear (2PN) and available embryos.” (Line 31) has a grammatical error.

-The sentence “Chian et al. [17] first demonstrated that a natural IVF cycle…” (Line 58) has grammatical errors and is not clear. Please rewrite it.

-“The number of good quality embryos was 2.2±2.5 IVM…” (Line 208)

- Please rewrite the sentence “Our data reinforce knowledge in this field, which currently has a limited number of studies on this 71 topic.” (Line 70)

Response: Thanks for your comments.

We have revised the text

It now reads:

- No statistically significant differences were observed in the number of 2 pronuclear (2PN) and available embryos.(line 40)

- Chian et al.[17] first demonstrated the pregnancies and live births that resulted from IVF of mature oocytes retrieved from dominant follicles in a natural cycle combined with IVM of immature oocytes retrieved from small follicles, and indicated that the natural cycle IVF/M had been an attractive choice for infertility treatment for various reasons.(line 70)

- The number of good quality embryos were 2.2±2.5 IVM vs. 2.1±2.3 in IVF/M (p=0.64). (line 212) 

- Our data made reinforcement knowledge in this field, which currently has a limited number of studies on this topic. (line 84)

*3. You need to consider the inclusion and exclusion criteria in more detail. If your PCOS patients were with UPOR, you should mention this point as well.

Response: We have rewritten the inclusion and exclusion criteria.

It reads:(line 96)

Infertile women diagnosed with PCOS according to the revised Rotterdam criteria (Rotterdam ESHRE/ASRM-Sponsored PCOS Consensus Workshop Group, 2004) scheduled for their IVF attempt were eligible to participate in the IVM. The advantages of informed consent are that all of the participants are getting a full explanation of the possible risks associated with the experiment, patients have begun receiving IVM treatments.

The inclusion criteria for the switching IVF to IVM group were designed to incorporate PCOS women whose characteristics indicated a high risk of developing unexpected poor ovarian responses (UPOR), for instance, in the COS process if follicular developmental retardation (only one to two mature follicles≥12mm or less) and more than 20 follicles (≤10mm) after being subjected to COH for 8~10 days.

The exclusion criteria were fertility preservation and preimplantation genetic testing cycles.

*4. There is a lot of heterogenicity in the study groups. As the results showed significant differences in age, BMI, method of fertilization, etc. How can you explain the effects of these factors on your results?

Response: Thanks for your question. (line 333)

The main limitations of our study include the retrospective of the IVF dataset and the availability of linked data between 2008 and 2017. There is a long timeframe that the participants have to be recruited. ART methodology and efficacy have significantly changed over the 9-year recruitment to the study and therefore, this may affect the result and conclusions drawn.

There is a lot of heterogenicity in the study groups, such as age, BMI, COH protocols, treatment, or clinic variables, similarly, there may be that may influence our results. Multiple regression analysis was used to analyze the factors that may affect the live birth rate after natural or switching to IVF/M. The potential confounders were chosen based on clinical experience and studies published in recent years, female age, BMI, COH protocols, and treatment.

After the stepwise logistic analysis, a number of factors were entered into the multivariable logistic regression (including androgen level, duration of the follicular phase, number of oocytes retrieved, and number of embryos transferred). However, only the number of embryos transferred was a significant independent factor associated with live birth after IVM on multivariate logistic regression analysis, which was consistent with existing literature. Further IVM protocol development and further RCTs are needed to evaluate the effectiveness and safety of IVM protocols.

*5. Keywords:Please revise according to alphabetical order.

Response: Thank you for your comment.

We have revised the text.

*6. Abbreviations:

All abbreviations should have been provided in full on the first mention and this applies to the title, abstract, main text, and each table/figure as they will be read independently.

Please explain these words when they appear for the first time:

OHSS in the abstract, AFC in line 290, and RCT in line 315. As you defined cLBR for the first time in line 86, please use this abbreviated form thereafter.

It is important to use uniform terminology in your work. For example, there is inconsistency in the PCOS definition as authors have used “polycystic ovary syndrome” in the abstract and “polycystic ovarian syndrome” in keywords, introduction, etc.

Please use FSH instead of “follicle-stimulating hormone” after explaining it for the first time (especially in Tables).

Response: Thank you for your comment.

We have revised the text.

*7. Tables:

Please redesign the tables as they are difficult to read and follow the contents. I suggest to align your contents in the first column with the left margin and use hanging for the subgroups.

The lower cases (a-g) and also the numbers (e.g. Kg/m2) used in tables should be written as superscripts.

Response: Thank you for your comment.

We have revised the text.

Table I:

It seems that the total percentage of some factors such as, “PCOS combined causes for infertility” does not reach 100%. You have not mentioned any missing data!!

Response: Thank you for your comment. (Table I)

PCOS is the primary infertility cause of 531 women in our study. There can be a single cause or multiple causes for female infertility. Furthermore, Women who had a combination of infertility factors: male factors, tubal infertility, endometriosis disorders, and previous poor ovarian response in IVF attempts were the most common combination seen. Single PCOS cause (33.5%); among the multiple causes of infertility, the male sex accounted for 36%, followed by tubal disorder (27.2%), previous POR (2.7%), and endometriosis (0.7%).

Table IV: As you mentioned about the missing number in the table footnote, please remove them from the Table. The definition of the β-hCG abbreviation is missing from the footnote.

Response: Thank you for your comment.We have revised the text.

*8. In line 188, you mentioned “Patients in the natural IVM group tended to have more previous IVF attempts than those in the switching IVF/M group” but your data in Table II does not support this statement. Could you please explain it?

Response: Thank you for your comment.

We have revised the text. It seems to be difficult to draw conclusions, the median results were both 1 attempt, the different was found in the value 75th percentile.

*9. The sentence “No significant difference was found between IVM and IVF/M cycles in terms of clinical pregnancy rate…” (Line 246) is not consistent with the results of clinical pregnancy in Table IV!!

Response: Thanks for your comments.

We have now corrected the text in this section.

Reviewer 4 Report

Thank you for submitting the article titled ‘Effectiveness, flexibility, and safety of switching IVF/M as a rescue strategy in unexpected poor ovarian response for PCOS infertility patients’ for review. 

The article is well structured and well written. It addresses and interesting aspect and aims to add more evidence to the field of IVM. The numbers of women included is substantial and therefore a strength of the study, as is REC approval. The use of cumulative birth rate as the outcome is the gold standard for reporting of fertility studies- it is good to see this in the paper. 

There are however some issues within the manuscript that will need to be addressed. These main limitations are:

1.     There is a long timeframe that the participants have been recruited over. ART methodology and efficacy has significantly changed over the 9 year recruitment to the study and therefore, this may affect the result and conclusions drawn. The authors need to acknowledge this limitation. Why is there no data included in the study after 2017?

2.     Please expand and describe the Rotterdam ESHRE/ ASRM criteria for PCOS, and reference appropriately (line 84). How did the change in the guidelines describing PCOS affect the retrospective inclusion of patients into the study? Please address and discuss. 

3.     There are many COH protocols included in the study increasing confusion and introducing heterogeneity into the study. The patients on different protocols should not be compared or included in the same group. Please discuss this as a limitation (line 100-102). Please describe in the methods how mini-stimulation cycles were performed. 

4.     Please reference the Istanbul consensus (line 121) and Gardner and Schoolcraft grading system (line 123).

5.     In the study populations women with <12 antral follicles were included. These are not considered polycystic ovaries then and therefore will behave differently during stimulation. Please explain why women with <12 follicles were included as PCOS patients? This is a substantial number of patients in the study (52 and 32 in the IVF/M and IVM groups, respectively) and poses the most significant limitation to the study. Subgroups should be used based on the AFC numbers and comparison of outcomes between these should be included. Otherwise, the results of the study are not generalisable. This may also be the reason why there is a substantial difference between the number of oocytes obtained between the two study groups. Please review. 

Author Response

SPECIFIC COMMENTS

Comments and Suggestions for Authors

Thank you for submitting the article titled ‘Effectiveness, flexibility, and safety of switching IVF/M as a rescue strategy in unexpected poor ovarian response for PCOS infertility patients’ for review.

The article is well structured and well written. It addresses and interesting aspect and aims to add more evidence to the field of IVM. The numbers of women included is substantial and therefore a strength of the study, as is REC approval. The use of cumulative birth rate as the outcome is the gold standard for reporting of fertility studies- it is good to see this in the paper.

Response: We appreciate your great help and the opportunity to revise our manuscript. 

There are however some issues within the manuscript that will need to be addressed. These main limitations are:

*1. There is a long timeframe that the participants have been recruited over. ART methodology and efficacy has significantly changed over the 9 year recruitment to the study and therefore, this may affect the result and conclusions drawn. The authors need to acknowledge this limitation. Why is there no data included in the study after 2017?

Response: Thanks for your question.(line 333)

The main limitations of our study include the retrospective of the IVF dataset and the availability of linked data between 2008 and 2017.

There is a lot of heterogenicity in the study groups, such as age, BMI, COH protocols, treatment, or clinic variables, similarly, there may be that may influence our results. Multiple regression analysis was used to analyze the factors that may affect the live birth rate after natural or switching to IVF/M. The potential confounders were chosen based on clinical experience and studies published in recent years, female age, BMI, COH protocols, and treatment.

After the stepwise logistic analysis, a number of factors were entered into the multivariable logistic regression (including androgen level, duration of the follicular phase, number of oocytes retrieved, and number of embryos transferred). However, only the number of embryos transferred was a significant independent factor associated with live birth after IVM on multivariate logistic regression analysis, which was consistent with existing literature. Further IVM protocol development and further RCTs are needed to evaluate the effectiveness and safety of IVM protocols.

The study was originally designed to continue through 2017 as there is always a 3-year lag in clinic reporting to capture live birth data for all cycles in the specified time frame. We completed data collection in late 2020.

*2. Please expand and describe the Rotterdam ESHRE/ ASRM criteria for PCOS, and reference appropriately (line 84). How did the change in the guidelines describing PCOS affect the retrospective inclusion of patients into the study? Please address and discuss.

Response: Thanks for your question.

We have supplemented the suggested references.

PCOS is a common endocrine disorder depending on the diagnostic criteria used, clinical manifestations of the syndrome are varied, and multiple parameters are needed for its diagnosis. The situation is aggravated by a plethora of definitions that professional societies and organizations use to diagnose PCOS. We found that most clinicians followed the Rotterdam 2004 criteria and others the latest ESHRE 2018 guidelines to diagnose PCOS. This is in line with studies performed in Europe and North America that showed most clinicians still used the Rotterdam criteria(oligo‐anovulation, hyperandrogenism and polycystic ovaries (≥ 12 follicles measuring 2‐9 mm in diameter and/or an ovarian volume > 10 mL in at least one ovary). On the other hand, a recent study found that the AE-PCOS criteria were most frequently used to diagnose PCOS in China.

In our study, Infertile women diagnosed with PCOS according to the revised Rotterdam criteria (Rotterdam ESHRE/ASRM-Sponsored PCOS Consensus Workshop Group, 2004) scheduled for their IVF attempt were eligible to participate in the study.

Another limitation of this study is the lack of clinical evaluation and metabolic parameters including those of AMH, androgens, triglycerides, high-density lipoproteins, and glucose tolerance. Furthermore, studies should be carried out among women with PCOS with different phenotypes to segregate the role of hyperandrogenism and metabolic derangements.

*3. There are many COH protocols included in the study increasing confusion and introducing heterogeneity into the study. The patients on different protocols should not be compared or included in the same group. Please discuss this as a limitation (line 100-102). Please describe in the methods how mini-stimulation cycles were performed.

Response: Thank you for your comment.

(1)Despite the advances in ARTs, one of the main challenges is the management of patients who have POR. Despite the high number of small follicles per ovary (2-3 times that of normal), there is poor follicular growth and development in response to gonadotropin stimulation. Researchers have proposed a series of strategies and ovarian stimulation protocols to improve pregnancy outcomes in patients with POR during their IVF/ICSI treatment. However, clinical decisions remain controversial. We did not further evaluate various COH protocols to account for the clinical outcomes and Case data collection is limited, with only dozens of cases in each group.

(2)Further research is needed to determine the etiology of UPOR, and identify markers that will allow us to reliably predict which patients for whom IVF is less appropriate than IVM.

(3)Minimal ovarian stimulation is started with an extended regimen (from day 3 of the cycle until the day before triggering) of CC /LE in conjunction with gonadotropin injections starting on days 4-7 of the cycle with 75-150 IU of human menopausal gonadotropins (hMG) daily. Patients usually receive both CC/LE and low-dose gonadotropins, and the dose given depends on the ovarian reserve status and the body mass index (BMI) of the patient. The final maturation of oocytes is usually induced by either intramuscular hCG or GnRH agonist (intramuscularly or nasally).

*4. Please reference the Istanbul consensus (line 121) and Gardner and Schoolcraft grading system (line 123).

Response: Thanks for your comment.

We have supplemented the suggested references.

*5. In the study populations women with <12 antral follicles were included. These are not considered polycystic ovaries then and therefore will behave differently during stimulation. Please explain why women with <12 follicles were included as PCOS patients? This is a substantial number of patients in the study (52 and 32 in the IVF/M and IVM groups, respectively) and poses the most significant limitation to the study. Subgroups should be used based on the AFC numbers and comparison of outcomes between these should be included. Otherwise, the results of the study are not generalisable. This may also be the reason why there is a substantial difference between the number of oocytes obtained between the two study groups. Please review.

Response: Thank you for your comment.

  • Infertile women diagnosed with PCOS according to the revised Rotterdam criteria (Rotterdam ESHRE/ASRM-Sponsored PCOS Consensus Workshop Group, 2004) scheduled for their IVF attempt were eligible to participate in the study. PCOS is defined by the presence of two of three of the following criteria: oligo‐anovulation, hyperandrogenism and polycystic ovaries (≥ 12 follicles measuring 2‐9 mm in diameter and/or an ovarian volume > 10 mL in at least one ovary).
  • Although some patients did not have polycystic ovaries, they are diagnosed with PCOS due to oligomenorrhea and hyperandrogenism. Following the 2003 Rotterdam criteria [13], PCOS can be divided into four different phenotypes: (A) hyperandrogenism, chronic anovulation, and polycystic ovaries (HA+AO+PCO); (B) chronic anovulation and polycystic ovaries but no clinical or biochemical hyperandrogenism (AO+PCO); (C) hyperandrogenism and chronic anovulation but normal ovaries (HA+AO); (D) hyperandrogenism and polycystic ovaries but ovulatory cycles (HA+PCO). Most research has focussed on the different PCOS phenotypes, the ovarian response to gonadotropin (Gn) is varied in controlled ovarian hyperstimulation, because of the small number of women in the investigated group and too short a period on the trial, this thesis needs to be the subject of further studies. Larger studies are needed to determine whether switching IVF to IVM is a widely applicable strategy for women who respond inappropriately to ovarian stimulation and its success rate and to the better recognition of its pathogenesis.

Round 2

Reviewer 3 Report

Thanks for revising the manuscript based on the comments. There are some minor points that were missed. Please consider these comments.

Please write the sentences as follows:

 ·       Line 313: “Previous studies reported that in patients who developed a severe form of the syndrome …”

·       Line 315: “This suggests that the small and medium (<14 mm) size follicles are mostly responsible …”

·       Line 210: “The number of good quality embryos was 2.2±2.5 in IVM vs. 2.1±2.3 in IVF/M group (p=0.64).”

·       Line 72: “Our data reinforce knowledge in this field, which currently has a limited number of studies.”   

·       Line 86: Please remove the sentence “The advantages of informed consent are that all of the participants are getting a full explanation of the possible risks associated with the experiment, patients have begun receiving IVM treatments.”

·       Line 89: “The inclusion criteria for the switching IVF to IVM group were designed to incorporate PCOS women whose characteristics indicated a high risk of developing unexpected poor ovarian responses (UPOR). For instance, in the COS process, UPOR is defined as follicular developmental retardation (only one to two mature follicles≥12mm or less) and more than 20 follicles (≤10mm) after being subjected to COH for 8~10 days.”

·       Line 240: Please use the abbreviated form of cLBR after its first definition in line 95. 

·       The abbreviation of human chorionic gonadotropin is β-hCG. Please insert it in the footnote of Table IV. 

Author Response

SPECIFIC COMMENTS

Comments and Suggestions for Authors

Thanks for revising the manuscript based on the comments. There are some minor points that were missed. Please consider these comments.

Response: We appreciate your great help and the opportunity to revise our manuscript. 

Please write the sentences as follows:

  •       Line 313: “Previous studies reported that in patients who developed a severe form of the syndrome …”
  •       Line 315: “This suggests that the small and medium (<14 mm) size follicles are mostly responsible …”
  •       Line 210: “The number of good quality embryos was 2.2±2.5 in IVM vs. 2.1±2.3 in IVF/M group (p=0.64).”
  •       Line 72: “Our data reinforce knowledge in this field, which currently has a limited number of studies.”   
  •       Line 86: Please remove the sentence “The advantages of informed consent are that all of the participants are getting a full explanation of the possible risks associated with the experiment, patients have begun receiving IVM treatments.”
  •       Line 89: “The inclusion criteria for the switching IVF to IVM group were designed to incorporate PCOS women whose characteristics indicated a high risk of developing unexpected poor ovarian responses (UPOR). For instance, in the COS process, UPOR is defined as follicular developmental retardation (only one to two mature follicles≥12mm or less) and more than 20 follicles (≤10mm) after being subjected to COH for 8~10 days.”
  •       Line 240: Please use the abbreviated form of cLBR after its first definition in line 95.
  •       The abbreviation of human chorionic gonadotropin is β-hCG. Please insert it in the footnote of Table IV.

Response: Thanks for your reminding. We have revised the text:

It now reads:

Line 313: Previous studies reported that in patients who developed a severe form of the syndrome, 95% of pre-ovulatory follicles were <16 mm and most were <9 mm.

Line 315: This suggests that the small and medium (<14 mm) size follicles are mostly responsible for the high serum oestradiol concentrations (and vasoactive compounds).

Line 210: The number of good-quality embryos was 2.2±2.5 in IVM vs. 2.1±2.3 in IVF/M group (p=0.64).

Line 72: Our data reinforce knowledge in this field, which currently has a limited number of studies.

Line 86: We’ve removed these as suggested.

Line 89: The inclusion criteria for the switching IVF to IVM group were designed to incorporate PCOS women whose characteristics indicated a high risk of developing unexpected poor ovarian responses (UPOR). For instance, in the COS process, UPOR is defined as follicular developmental retardation (only one to two mature follicles≥12mm or less) and more than 20 follicles (≤10mm) after being subjected to COH for 8~10 days.

Line 240: We have revised the text.

Table IV: We’ve added this as suggested.
